# Near-Infrared Spectroscopy Monitoring in Cardiac and Noncardiac Surgery: Pairwise and Network Meta-Analyses

**DOI:** 10.3390/jcm8122208

**Published:** 2019-12-14

**Authors:** Christian Ortega-Loubon, Francisco Herrera-Gómez, Coralina Bernuy-Guevara, Pablo Jorge-Monjas, Carlos Ochoa-Sangrador, Juan Bustamante-Munguira, Eduardo Tamayo, F. Javier Álvarez

**Affiliations:** 1Department of Cardiac Surgery, University Clinical Hospital of Valladolid, Ramon y Cajal Ave. 3, 47003 Valladolid, Spain; christlord26@gmail.com (C.O.-L.); jbustamantemunguira@gmail.com (J.B.-M.); 2BioCritic. Group for Biomedical Research in Critical Care Medicine, Ramon y Cajal Ave. 7, 47005 Valladolid, Spain; pablojor@yahoo.es (P.J.-M.); tamayo@med.uva.es (E.T.); alvarez@med.uva.es (F.J.Á.); 3Pharmacological Big Data Laboratory, Department of Pharmacology and Therapeutics, University of Valladolid, Ramon y Cajal Ave. 7, 47005 Valladolid, Spain; coralber@gmail.com; 4Department of Anatomy and Radiology, Faculty of Medicine, University of Valladolid, Ramon y Cajal Ave. 7, 47005 Valladolid, Spain; 5Department of Anaesthesiology, University Clinical Hospital of Valladolid, Ramon y Cajal Ave. 3, 47003 Valladolid, Spain; 6Department of Surgery, Faculty of Medicine, University of Valladolid, Ramon y Cajal Ave. 7, 47005 Valladolid, Spain; 7Clinical Epidemiology Support Office, Sanidad Castilla y León, Requejo Ave. 35, 49022 Zamora, Spain; cochoas2@gmail.com; 8Ethics Committee of Drug Research–East Valladolid, University Clinical Hospital of Valladolid, Ramon y Cajal Ave. 3, 47003 Valladolid, Spain

**Keywords:** bSo_2_, NIRS, monitoring, near-infrared spectroscopy, cardiac surgery, perioperative care

## Abstract

Goal-directed therapy based on brain-oxygen saturation (bSo_2_) is controversial and hotly debated. While meta-analyses of aggregated data have shown no clinical benefit for brain near-infrared spectroscopy (NIRS)-based interventions after cardiac surgery, no network meta-analyses involving both major cardiac and noncardiac procedures have yet been undertaken. Randomized controlled trials involving NIRS monitoring in both major cardiac and noncardiac surgery were included. Aggregate-level data summary estimates of critical outcomes (postoperative cognitive decline (POCD)/postoperative delirium (POD), acute kidney injury, cardiovascular events, bleeding/need for transfusion, and postoperative mortality) were obtained. NIRS was only associated with protection against POCD/POD in cardiac surgery patients (pooled odds ratio (OR)/95% confidence interval (CI)/I^2^/number of studies (n): 0.34/0.14–0.85/75%/7), although a favorable effect was observed in the analysis, including both cardiac and noncardiac procedures. However, the benefit of the use of NIRS monitoring was undetectable in Bayesian network meta-analysis, although maintaining bSo_2_ > 80% of the baseline appeared to have the most pronounced impact. Evidence was imprecise regarding acute kidney injury, cardiovascular events, bleeding/need for transfusion, and postoperative mortality. There is evidence that brain NIRS-based algorithms are effective in preventing POCD/POD in cardiac surgery, but not in major noncardiac surgery. However, the specific target bSo_2_ threshold has yet to be determined.

## 1. Introduction

Serious repercussions of cardiac and noncardiac procedures, including acute kidney injury (AKI) [1,2,3], respiratory complications [4], postsurgical bleeding [5], and major adverse cardiovascular events (MACE), such as stroke [6] and acute myocardial infarction [7], are still common [8]. Neurological events, including postoperative cognitive decline (POCD) and postoperative delirium (POD), are some of the most important complications [9,10], increasing healthcare costs by 40% of these patients [11] and enormously impairing their quality of life [6,12].

The etiology of these complications is multifactorial and poorly understood. Thus, patients need to be closely monitored [13]. Near-infrared spectroscopy (NIRS) is a promising technique that continuously maintains oxygen saturation within a regional tissue area [14]. Despite the cerebral benefits of brain-oxygen-saturation (bSo_2_) monitoring [15], whether NIRS should be a standard of care remains hotly debated [16,17]. It is not yet routine [18] and is still regarded as an insufferable parameter to monitor by anesthesiologists [19]. The central nervous system is rarely directly monitored and is often managed through indirect variables [9].

The scarcity of evidence regarding the clinical benefits of NIRS leads to uncertainty in its use. We hypothesize that bSo_2_ monitoring may reduce the incidence of major postoperative adverse outcomes, including POCD/POD, MACEs, AKI, bleeding/need for transfusion, and mortality, in patients undergoing cardiac and noncardiac surgery. To tackle this controversy, taking into account the fact that different NIRS target values have never been compared head to head in either randomized controlled trials (RCTs) or currently available meta-analyses, we conducted conventional pairwise and network meta-analyses to evaluate the effectiveness of individualized NIRS-based measures in decreasing the occurrence of these clinical outcomes.

## 2. Materials and Methods

### 2.1. Systematic Review Process

Internet database-specific search strategies were developed and applied to MEDLINE, EMBASE, and the Cochrane Controlled Register of Trials (CENTRAL), as well as study registries, repositories, and meeting abstracts archives up to April 2019 (Appendix A). According to study eligibility (Table 1), 1 reviewer initially scanned the results to exclude clearly irrelevant studies. Thereafter, 2 reviewers independently evaluated titles, abstracts, and full-text reports (conciliation and corresponding author contact were planned).

### 2.2. Statistical Analysis

After assessing study quality [20], and before applying planned contextualization techniques [21,22,23], pairwise and network meta-analyses of cardiac- and noncardiac-surgery-patient aggregate-level data were performed. The overall odds ratio (OR) and 95% confidence interval (CI) were obtained for various postoperative outcomes corresponding to the use of NIRS monitoring (POCD/POD, MACEs, AKI, bleeding/need for transfusion, and mortality) by the Mantel–Haenszel random-effects method. The heterogeneity and presence of reporting bias was verified using Review Manager (RevMan) software version 5.3 (Cochrane Collaboration, Copenhagen, Denmark). Pooled ORs, with their corresponding 95% credible intervals (95% CrIs) for the aforementioned outcomes, were also calculated via Bayesian network meta-analysis (Markov chain Monte Carlo simulation on vague priors using a random-effects model for ‘bad’ outcomes and 0 values correction). They were presented in league tables with a calculation of the surface under the cumulative ranking area (SUCRA) for various competing NIRS target values (bSo_2_ > 70%/75%/80%/85%, and nondrop bSo_2_ > 15%/20% from baseline), after verifying convergence and inconsistency using NetMetaXL software (Canadian Agency for Drugs and Technologies in Health, Ottawa, Canada, and Cornerstone Research Group, Ontario, Canada) [24]. The Grades of Recommendation, Assessment, Development, and Evaluation (GRADE) approach was used for rating the confidence of the obtained effect estimates [20].

## 3. Results

Results of the pairwise and network meta-analyses were presented according to the Preferred Reporting Items for Systematic reviews and Meta-Analyses (PRISMA) extension statement for the reporting of systematic reviews that incorporated network meta-analyses of healthcare interventions [25] and that followed a registered protocol in the International Prospective Register of Systematic Reviews (PROSPERO; registration ID: CRD42019131417). This ensured study suitability in terms of nonredundancy, transparency, and adequacy of the chosen methods (prospective recording of the planned review details) [26].

### 3.1. Study Characteristics

After the search process, step-by-step results, which are presented graphically in the standard flowchart provided by the PRISMA group (Figure 1) [27], a total of 12 RCTs were eligible [28,29,30,31,32,33,34,35,36,37,38,39]. They involved a total population of 1626 major surgery patients (cardiac and noncardiac surgical procedures). The details of the participants, interventions, comparators, main and secondary outcomes, and key features of eligible studies (RCT design details, follow-up data) are available in tabular form (Appendix A). The same information is also accessible online in tabular form for three eligible RCTs [19,40,41] that were excluded because of a lack of numerical data for the dichotomous analysis of the planned outcome (Appendix A). Overall, all of these studies presented evidence for the use of NIRS in the last 15 years and involved clinical activity in developing and developed countries from all over the world. All of these studies were reported as peer-reviewed journal articles in highly ranked Journal Citation Reports (JCR) journals. No unpublished reports corresponding to these studies were found, nor were other unpublished studies available.

Other excluded records consisted of observational studies, case series of patients who underwent major surgery and opinion, or narrative review articles focused particularly on the cardiac-surgery-patient population. Interestingly, on average, 10% of the reports presenting noninterventional data were available as unpublished reports (oral and posted abstracts presented in relevant international meetings).

According to the within-study risk of bias assessment, the methodological quality of eligible studies was considered moderate to high (Appendix A). Importantly, POCD/POD was reported with a high lack of harmony (i.e., various scoring systems were used), and with the rationale that it would not be clinically relevant. However, when separately examining the articles, all of the studies suggested a relationship between the use of NIRS monitoring and improved outcomes in study participants who did not undergo the comparators. The risk of selective reporting bias was high when the sample size was taken into account.

### 3.2. Outcomes

In the pairwise meta-analyses of cardiac- and noncardiac-surgery patients (all cardiac-surgery patients underwent cardiopulmonary bypass), in seven and three of the 12 eligible RCTs, respectively (Figure 2), the pooled ORs and 95% CIs for the outcome of POCD/POD were 0.34 (0.14 to 0.85) and 0.33 (0.10 to 1.14). Overall, for cardiac- and noncardiac-surgery patients, the OR and 95% CI for this outcome was in favor of intervention, at 0.34 (0.17 to 0.67). Importantly, no heterogeneity in subgroup comparisons, and critical funnel-plot asymmetry affecting these summary estimates, was observed. In addition, the benefit of the use of NIRS monitoring was undetectable in the Bayesian network meta-analysis. No superiority of any NIRS target values in any of the 10 trials included in analysis was noted (Figure 3). Interestingly, on the basis of SUCRA, with respect to other target values, maintaining bSo_2_ > 80% of baseline appeared to have the most pronounced impact on the outcome of POCD/POD. This was true even when considering the influence of heterogeneity (vague priors) on network-diagram geometry (Figure 4A), which was counterbalanced by the absence of inconsistency (Figure 4B).

According to GRADE, effect estimates for this outcome should be considered as derived from a low-quality body of evidence. The quality rating fell by two levels for heterogeneity and risk of reporting bias, even considering that there was no influence of indirectness in participants, interventions, comparators, outcomes, or imprecision in summary estimates (acceptably wide CIs except in the subgroup of noncardiac-surgery patients).

In the pairwise meta-analysis of five, six, six, and two trials out of the 12 eligible RCTs, the pooled OR and 95% CI for outcomes of postoperative mortality, MACEs, AKI, and bleeding/need for transfusion were 0.58 (0.22 to 1.50), 0.47 (0.20 to 1.10), 0.69 (0.43 to 1.12), and 0.43 (0.10 to 1.94), respectively. Interestingly, when analyzing data on morbidity (combination of POCD/POD, MACEs, and AKI), the OR and 95% CI were 0.41 (0.25 to 0.70). These calculations were not affected by heterogeneity (I^2^ = 0%, *p* > 0.1), with the exception of bleeding/need for transfusion (I^2^ = 82%, *p* = 0.02) and the combined outcome of morbidity (I^2^ = 67%, *p* = 0.0004). However, funnel-plot asymmetry was evident (Appendix A). Remarkably, in Bayesian network meta-analysis, maintaining a higher bSo_2_ (>75–80%) relative to baseline, rather than avoiding a drop in bSo_2_, appeared to have the most pronounced impact on outcomes of postoperative mortality, MACEs, and AKI according to the SUCRA-based ranking (Appendix A). This was true even when considering heterogeneity that was compensated for by freedom from inconsistency (Appendix A).

With the exception of bleeding/need for transfusion, effect estimates for the remaining outcomes should be considered as derived from a low-quality body of evidence according to GRADE. This is taking into account the imprecision of pooled estimates (wide CIs) and risk of reporting bias. Heterogeneity, risk of reporting bias, and imprecision downgraded the quality for the outcome of bleeding/need for transfusion to very low.

## 4. Discussion

### 4.1. Major Findings

This systematic review and meta-analysis demonstrated reductions in POCD/POD, attributable to the use of individualized NIRS-based interventions only during major cardiac surgery and not major noncardiac surgery. While maintaining bSo_2_ > 80% of the baseline appeared to be a safety threshold, the specific bSo_2_ threshold resulting in the greatest clinical benefit is yet to be determined. In addition, the use of NIRS monitoring did not result in an occurrence reduction of AKI, cardiovascular events, bleeding/need for transfusion, or postoperative mortality.

### 4.2. Strengths and Limitations

This is the first summary of evidence applying and presenting the findings of analysis allowing for the comparison of methodologically independent exposures (i.e., several NIRS target values) that have not been previously assessed for critical outcomes in the context of major surgery. To date, only conventional pairwise meta-analysis techniques and qualitative synthesis of skewed data have been performed [11,42]. Conventional pairwise meta-analysis is insufficient when many options for tools (interventions) exist (e.g., when several drugs are available for the same treatment strategy). Network meta-analysis provides a viable solution, even though some interventions/exposures have never been compared head to head in RCTs [21]. Furthermore, to the best of our knowledge, this is the first contemporary review of NIRS monitoring in both types of major procedures.

Although the comprehensiveness of searches is guaranteed, the risk of reporting bias (funnel-plot asymmetry) is more of a concern than the mathematical confirmation of missing more imprecise studies (probably unpublished) in analyses [43,44]. Reporting bias can lead to overly optimistic conclusions in systematic reviews [45]. Likewise, statistical heterogeneity, mainly related to inherent differences between major cardiac- and noncardiac-surgery patients, reduces the generalizability of the findings of this meta-analysis [46]. This is true even for subgroup analyses that were performed [47]. There were no trials in pediatric cardiac surgery, which is a common setting for the use of this technology. Thus, relevant information was missed regarding the benefits of this monitoring method in the pediatric population. Furthermore, while POCD/POD may be part of a continuum, they may have different implications. Thus, they may be considered separately.

### 4.3. Clinical Implications

POD has been shown to be associated with a longer and costlier hospital course, a higher likelihood of death within six months, and postoperative institutionalization. Similarly, POCD has been related to increased mortality, poor quality of life, higher risk of prematurely leaving the work force, and increased dependence on social transfer payments [48].

Moreover, POCD/POD is especially common in elderly patients, given that advanced age is an acknowledged risk factor for its development. This bears severe and long-lasting consequences for these patients [49]. With aging global populations, an increasing number of both cardiac and noncardiac procedures, and scarcity in available therapeutic alternatives for POCD/POD, timely measures directed towards its prevention remain an issue of paramount importance [50]. In this regard, Lei et al. reported that low preoperative bSo_2_ may be related to an increased frequency of POD after cardiac surgery in patients >60 years old [51]. Similarly, Radtke and colleagues showed that intraoperative neuromonitoring was associated with lower rates of POD in high-risk surgical patients [52].

The results of this systematic review not only support the hypothesis that cerebral NIRS-based algorithms can prevent POCD/POD, but also substantially strengthen the clinical basis for the use of NIRS monitoring in cardiac surgery. This work supports a large number of reports that revealed NIRS as an early detector of regional oxygen metabolism imbalance [30,33,34,41,53].

This technology has also been used in other settings, such as discrimination between stable and unstable atherosclerotic plaques [54], the diagnosis and treatment of depression, schizophrenia, and Alzheimer’s disease [55], and the assessment of tissue perfusion in neonates [56]. Subsequent actions performed by already standardized algorithms to prevent bSo_2_ desaturation could prevent long-lasting and devastating complications [30,33,36,40]. Reductions in NIRS values have been used to promptly identify devastating cerebral events, while more invasive monitoring metrics remain unchanged [19,57,58].

Although bSo_2_ monitoring is routinely performed in many cardiac-surgery centers, it is not a universally standardized practice, and has recently been used in specific major noncardiac surgeries [9,29,31]. In orthopedic surgery, for instance, this technology has been exclusively used in patients undergoing shoulder surgery in the beach-chair position [32,59,60]. This could be a reason why there is insufficient evidence to support its use in this type of procedure. Only four out of 12 RCTs included in this meta-analysis examined the effects of NIRS-based interventions in noncardiac surgery.

Similar to this study, Chan et al. performed meta-analysis and identified that reference bSo_2_ baseline values in cardiac surgery widely vary [61]. They found that preoperative bSo_2_ values are quite heterogeneous, and that there is a lack of knowledge of postoperative changes in bSo_2_. This might explain the considerable heterogeneity among general NIRS data reported in studies [62]. Indeed, each RCT arbitrarily defined their specific NIRS-based algorithms, making analysis highly complex, and significantly reducing the chance of identifying the target bSo_2_ threshold with the greatest clinical value in terms of lowering the risk of POCD/POD, AKI, and postoperative mortality. Nonetheless, maintaining bSo_2_ > 80% of the baseline seems to yield the best results.

It is clear that, without feedback from a specific indicator of end-organ damage, i.e., bSo_2_, the capacity to detect and enhance otherwise silent but potentially catastrophic perturbations with clinical variables remains significantly restricted [36]. Thus, NIRS monitors should be considered an essential component of generally accepted intraoperative monitoring in cardiac surgery. The implementation of such a simple measure, for instance, as a further essential requirement of the World Health Organization Safer Surgery checklist before and during surgery, may minimize the occurrence of adverse outcomes [63]. Adaptation of the checklist might help to avoid complications with devastating consequences such as POCD.

The capacity of brain-injury biomarkers to predict or prevent POCD/POD in a timely manner is considerably restricted, as they are relatively nonspecific and inconclusive. Neuron-specific enolase (NSE) is an intracellular enzyme found in neurons, neuroendocrine cells, platelets, and erythrocytes [64]. S-100B is expressed in adipose tissue and the thymus [65]. An initial elevation in S-100B levels due to brain injury occurs at least 24 h after surgery [65]. S-100 release is biphasic, starting three to five hours after brain injury and occurring a second time 48 h after the insult [66]. S-100B shows a similar pattern. The peak level of S-100B occurs on the third day following a stroke. The cerebrospinal fluid levels of S-100 and S-100B are more accurate than the serum levels of S-100 and S-100B for evaluating cerebral damage [66]. However, these samples are limited [65]. Furthermore, they cannot be continuously assessed [28,37,67], and generally require a substantial amount of money to assess [64]. Thus, no optimal markers have emerged with regards to specificity or predictive value [68].

Conversely, the results presented here do not support the hypothesis that the use of bSo_2_ monitoring, using the brain as the index organ, reduces injury to the heart, kidneys, or lungs, or reduces mortality as a result of improved overall perfusion during cardiac or noncardiac surgery. This is similar to the conclusions of other authors [37,38,42]. This could be explained by cerebral autoregulation, which ensures relatively constant brain blood flow, even in cases of poor systemic perfusion. Olbrecht and colleagues showed that bSo_2_ remains unchanged in low-arterial-pressure events [69]. Furthermore, Chan et al. detected a lack of evidence for the correlation between bSo_2_ changes and postoperative outcomes because almost all studies have only focused on the intraoperative stage. This has left an unexplored gap in the postoperative period, where NIRS dynamics is also vital and may help to further understand clinical consequences [61]. Ortega-Loubon et al. demonstrated that postoperative kidney oxygen saturation (kSo_2_), but neither intraoperative kSo_2_ nor bSo_2,_ was correlated with the occurrence of postoperative AKI [14,70]. Therefore, it may be necessary to more closely monitor susceptible organs that are more likely to suffer acute injury during and after major procedures.

## 5. Conclusions

Existing evidence suggests that the use of patient-specific NIRS-based algorithms that aim to maintain bSo_2_ during cardiac surgery result in a reduction in the occurrence of POCD/POD, but not AKI, cardiovascular events, bleeding/need for transfusion, or postoperative mortality. Aiming to maintain bSo_2_ > 80% of the baseline seems to yield the maximum benefit in terms of preventing POCD/POD in cardiac surgery. This analysis contributes to the debate regarding the use of NIRS monitoring, at least in cardiac surgery. To address unanswered questions, further clinical trials of countermeasures against adverse outcomes after major noncardiac procedures are needed.

## Figures and Tables

**Figure 1 jcm-08-02208-f001:**
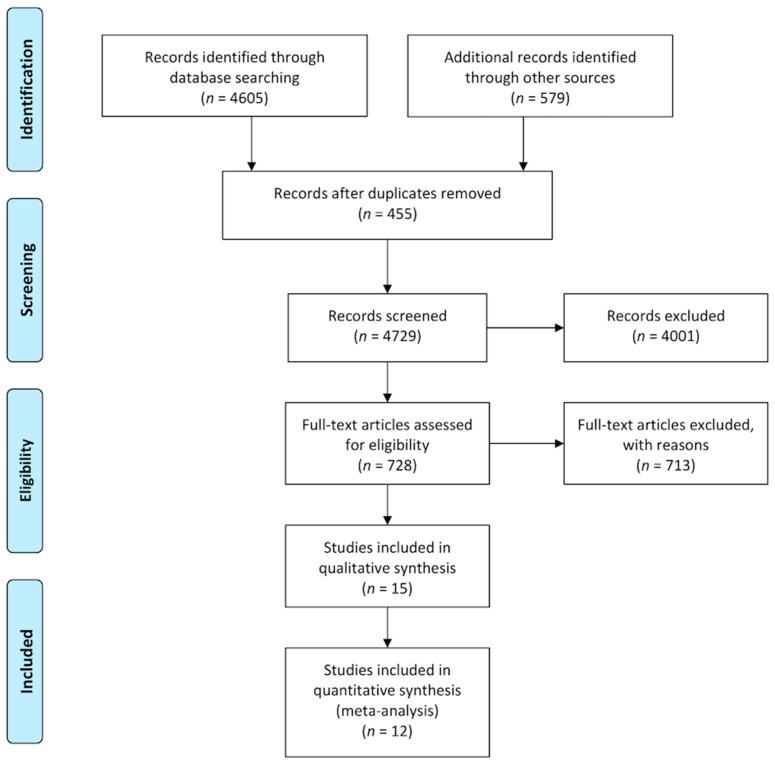
Preferred Reporting Items for Systematic Reviews and Meta-Analyses (PRISMA) flowchart representing systematic-review selection process.

**Figure 2 jcm-08-02208-f002:**
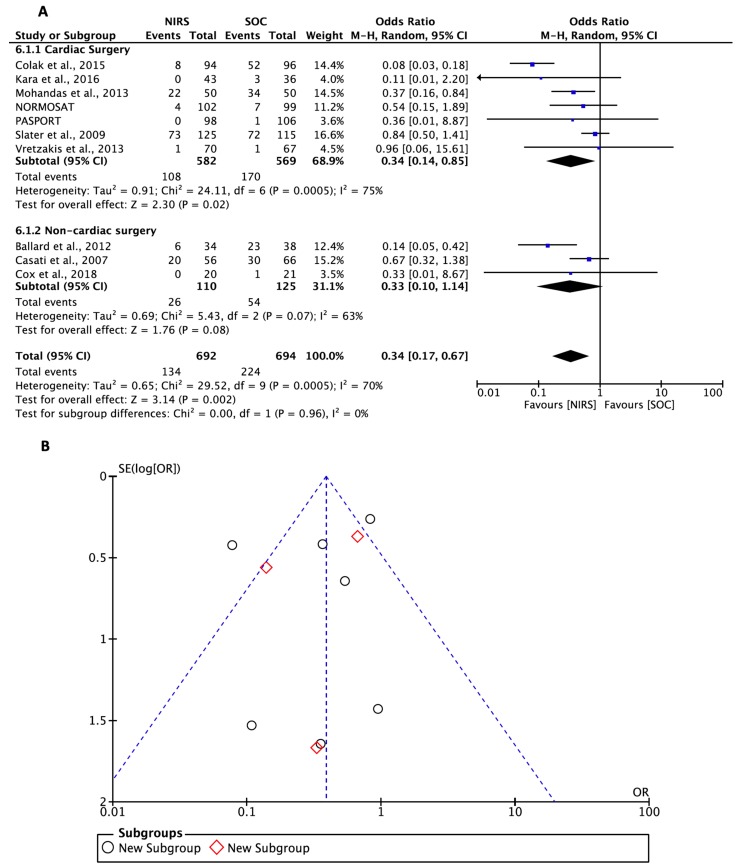
Effect of NIRS monitoring on postoperative cognitive decline/delirium. (**A**) Forest plot and (**B**) funnel plot. CI, confidence interval; M–H, Mantel–Haenszel test; NORMOSAT, normal cerebral oxygen saturation; PASPORT, Patient-Specific Cerebral Oxygenation Monitoring as Part of an Algorithm to Reduce Transfusion during Heart Valve Surgery: A Randomized Controlled Trial; SE, standard error; and SOC, standard of care.

**Figure 3 jcm-08-02208-f003:**
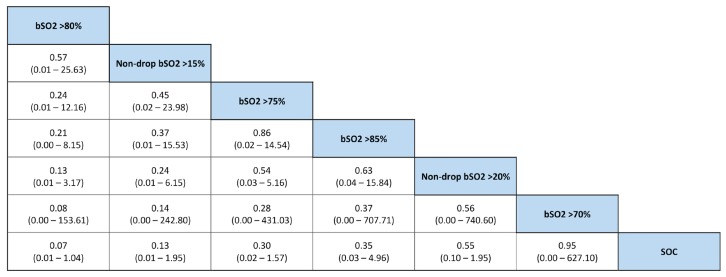
League table presenting all assessable NIRS target values for outcome of postoperative cognitive decline/delirium. bSo_2,_ brain-oxygen saturation.

**Figure 4 jcm-08-02208-f004:**
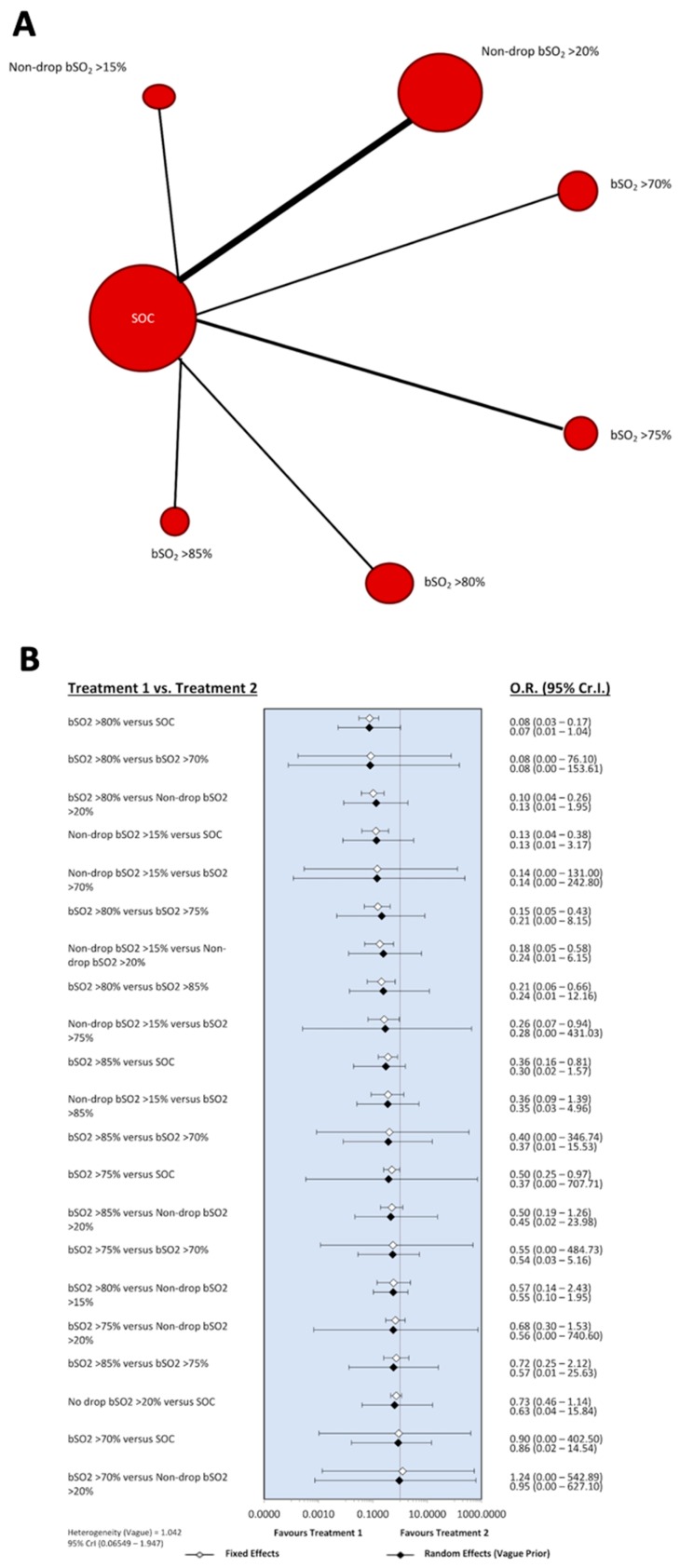
Network diagram and forest plot of competing NIRS target values for outcome of postoperative cognitive decline/delirium. CrI, credible interval; OR, odds ratio.

**Table 1 jcm-08-02208-t001:** Review questions and study eligibility.

	Explanations
Review question	Which perioperative NIRS target values were most adequate for patients who underwent major surgical procedures?
Participants/population	Adult and pediatric individuals who underwent major cardiac or noncardiac surgery.
Intervention(s)/exposures(s)	Perioperative near-infrared spectroscopy target values according to current algorithms.
Comparators	No brain-oxygenation monitoring or brain-oxygenation monitoring based on non-NIRS technologies.

Note: NIRS, near-infrared spectroscopy.

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
