# Peer review of "Near-Infrared Spectroscopy Monitoring in Cardiac and Noncardiac Surgery: Pairwise and Network Meta-Analyses"

_jcm, 2019, doi:10.3390/jcm8122208_

Round 1
Reviewer 1 Report
I have no further comments
Reviewer 2 Report
All of the reviewer's comments were taken into account and incorporated into the manuscript. I wish the authors all the best for their future endeavors.
This manuscript is a resubmission of an earlier submission. The following is a list of the peer review reports and author responses from that submission.
Round 1
Reviewer 1 Report
Overall well-done study using network meta-analysis. The authors did not overstate the significance of their findings and came to an appropriate conclusion. I would add, however, that not all cardiac surgery is the same. The articles used in this analysis for NIRS use in cardiac surgery, all involved the use of cardiopulmonary bypass (e-Table 1). This ought to be clearly stated in the article as this is a significant difference between the abdominal, orthopedic operations and the cardiac surgery operations.
Reviewer 2 Report
This is an interesting paper on NIRS advantages in cardiac and non-cardiac surgery patients. Analysis is performed appropriately and study reporting is performed using international standards. I have only minor objections:
Figure 2 should be presented with more caution: the reader may be confused about its description, i.e. two “sub-groups”. Please explain which is cardiac, and which is non-cardiac. Figure 3 needs to be explained in more details. This type of statistical method is used less frequently and the figure is difficult to understand. More to the point, this is rather a table than a figure. Please consider performing additional sub-analysis using “morbidity” as an outcome. This category may include all organ-perfusion related complications (i.e. kidney-related, neurological, cardiovascular). It would be interesting to assess if there is any advantage of using NIRS in cardiac and non-cardiac surgery to prevent perioperative complications related to organ hypoperfusion.Reviewer 3 Report
Dear Editor-in-Chief,
Thank you for giving me the opportunity to review the manuscript titled: “Near-Infrared Spectroscopy Monitoring in Cardiac and Non-Cardiac Surgery: A Pairwise Network Meta-Analysis”.The problem investigated by the authors is very important. Their efforts should be appreciated, because the optimal threshold for NIRS monitoring during major surgery remains to be elucidated. The manuscript is written well, yet there are some points that need improvement or clarification.
Abstract: It is well written and adapted to the PRISMA guidelines. The parts included in the abstract are adequate. The following changes are necessary:
Line 31: Please do not confuse the two abbreviations: POCD and POD. POCD stands for Postoperative Cognitive Decline and POD stands for Postoperative Delirium. The authors use it through the whole manuscript. Please change “POD/delirium” to POCD/POD.
Introduction: It is well-written, minor changes are necessary:
Line 48: Please add POD for Postoperative Delirium and use the term throughout the whole manuscript.
Line 50: Please underline the problem of POCD and POD in elderly patients, maybe add a short paragraph and consider the following references:
1. Kotfis K, Szylińska A, Listewnik M, et al. Early delirium after cardiac surgery: an analysis of incidence and risk factors in elderly (≥65 years) and very elderly (≥80 years) patients. Clin Interv Aging. 2018;13:1061–1070. Published 2018 May 30. doi:10.2147/CIA.S166909
2. Choi JY, Kim KI, Kang MG, et al. Impact of a delirium prevention project among older hospitalized patients who underwent orthopedic surgery: a retrospective cohort study. BMC Geriatr. 2019;19(1):289. Published 2019 Oct 26. doi:10.1186/s12877-019-1303-z
3. Radtke FM, Franck M, Lendner J, Kruger S, Wernecke KD, Spies CD. Monitoring depth of anaesthesia in a randomized trial decreases the rate of postoperative delirium but not postoperative cognitive dysfunction. Br J Anaesth. 2013 Jun;110 Suppl 1:i98-105. doi: 10.1093/bja/aet055. Epub 2013 Mar 28.
Line 55: Please re-phrase, “disgusting” monitor is not a very medical term.
Line 60: Here POCD/delirium abbreviation is used. Please do not confuse the two abbreviations: POCD and POD. POCD stands for Postoperative Cognitive Decline and POD stands for Postoperative Delirium (see remarks for the Abstract).
Materials and Methods: Generally, there is a clear explanation of the study, data collection and analysis, although the reviewer one concern - Figure 1: Please indicate the correct number of studies in each box in the PRISMA Flowchart. The numbers do not add up.
Results: This section is adequately described and clear.
Discussion: The authors discuss their results adequately. They emphasize the importance of the results obtained by their analysis and further discuss it with the most recent literature.
Conclusion: The results support the conclusions drawn from the study.
References: There are many current references that justify the study.
This research paper is valuable to the readership.The manuscript needs minor punctuation, spelling and style improvement.
With best regards
Reviewer 4 Report
In this study the authors have performed a network meta-analysis involving both cardiac and major non cardiac procedures to evaluate the usefulness of NIRS in improving various outcomes.
This is an important issue to be studied. However, I have some comments.
Page 2: Introduction line 56: What do you mean by a disgusting monitor by anesthesiologists. Ref 19 to which you refer has not much to do with this statement. Please revise this.
Page 7 Discussion: The statement "Although bSO2 monitoring is currently a standard of care in cardiac surgery" is not right. It is not a standard of care. It is routinely performed in many centers. This is what needs to be mentioned.
Page 8 Discussion line 237: The authors comment on the fact that NIRS should be used as a crucial component of the generally accepted intraoperative monitoring. This is a personal comment and is not based on your obtained results. Please revise.
Page 8 Discussion: Why do the authors discuss neurobiomarkers as pathophysiological mechanism of POCD. This is out of the scope of this manuscript.